# Nano formulation development and antibacterial activity of cinnamon bark extract-chitosan composites against *Burkholderia Glumae* the causative agent of *Bacterial Panicle Blight* in rice

Qamar Mohammed-Naji[1,2☯], Dzarifah Zulperi[1,3☯]*, Khairulmazmi Ahmad[1☯], Erneeza Mohd Hata[4☯]

1 Department of Plant Protection, Faculty of Agriculture, Universiti Putra Malaysia, Serdang, Selangor, Malaysia, 2 Directorate of the Planting of Holy Karbala, Iraqi Ministry of Agriculture, Baghdad, Iraq, 3 Laboratory of Sustainable Resources Management, Institute of Tropical Forestry and Forest Products, Universiti Putra Malaysia, Serdang, Selangor, Malaysia, 4 Institute of Plantation Studies, Universiti Putra Malaysia, Serdang, Selangor, Malaysia

☯ These authors contributed equally to this work.
* dzarifah@upm.edu.my

## Abstract

Bacterial panicle blight (BPB) disease, caused by *Burkholderia glumae*, poses a significant threat to rice production. Conventional chemical control methods contribute to environmental concerns and resistance issues, necessitating the development of sustainable alternatives. This study aimed to formulate and evaluate cinnamon bark extract-chitosan (CBE-CS) nano formulations for antibacterial efficacy against *Burkholderia glumae*. First, the antibacterial activity of cinnamon bark extract (CBE) was assessed, revealing a minimum inhibitory concentration (MIC) of 6.25 µg/mL and a minimum bactericidal concentration (MBC) of 12.5 µg/mL. Morphological analysis using scanning electron microscopy (SEM), confocal laser scanning microscopy (CLSM), and transmission electron microscopy (TEM) showed significant bacterial cell wall damage, cytoplasmic leakage, and structural degradation after treatment. Chemical characterization of CBE using gas chromatography-mass spectrometry (GC-MS) and Fourier transform infrared spectroscopy (FTIR) identified key active compounds, with (Z)-3-phenylacryldehyde as the major component (51.24%). Next, nano formulations of CBE-CS were developed, and their physicochemical properties were characterized, including particle size, zeta potential, encapsulation efficiency (33.9%), and loading capacity (48.78%). Antibacterial assessments demonstrated that the nano formulations effectively inhibited *Burkholderia. glumae*. Finally, greenhouse trials on rice seedlings confirmed the efficacy of these nano formulations in controlling BPB disease, showing significant bacterial suppression and improved plant health. These findings suggest that CBE-CS nanoparticles offer a promising, eco-friendly alternative for managing bacterial blight in rice, providing both effective antibacterial activity and enhanced plant protection.

**Data availability statement:** All relevant data are within the manuscript and its Supporting Information files.

**Funding:** The author(s) received no specific funding for this work.

**Competing interests:** The authors have declared that no competing interests exist.

## 1. Introduction

Rice is a primary dietary staple for many nations, with Malaysia being particularly reliant on it for food security [1]. However, rice cultivation faces significant challenges from various agricultural diseases and pests, among which *Burkholderia. glumae* (*B. glumae*) poses a severe threat [2]. *B. glumae* is the causative agent of bacterial panicle blight (BPB), a disease that spreads through rice seeds and can result in yield losses of up to 75% in affected fields. This pathogen is characterized by its rod shape, Gram-negative nature, polar flagella, and lack of spores, making it a persistent menace to rice crops [3]. Infected rice plants exhibit symptoms such as grain and leaf discoloration, cluster blight, mold spots on seeds, and a reduction in spikelet yield [2]. This bacterium produces toxic substances, particularly *toxoflavin*, a yellow pigment essential for pathogenicity that induces rot in rice seedlings. The disease spreads rapidly in tropical climates, facilitated by moderate temperatures, frequent rainfall, and high humidity during the rice panicle appearance [4]. Earlier findings to control *B. glumae* have explored various strategies, with plant extracts emerging as a promising alternative. These extracts have shown high efficacy in enhancing seed resistance and disrupting infection pathways, offering a potential solution to the limitations of traditional chemical treatments [5] and [6].

The cinnamon plant (*Cinnamomum zeylanicum Blume*), a member of the *Lauraceae* family, has been widely utilized in various scientific fields for its exceptional antimicrobial properties and abundance of essential natural compounds [7]. Cultivated in many countries, the bark of the cinnamon tree is harvested to produce cinnamon spice, which is commonly used in the preparation of various foods, including chocolates, beverages, sweets, and alcoholic drinks. Additionally, cinnamon is a key ingredient in the production of essential oils for cosmetic and food products [8] and [9]. Cinnamon's efficacy against microbial diseases is attributed to its rich phytochemical composition, which includes compounds such as cinnamaldehyde*, eugenol,* and *cinnamic acid* derivatives [10]. Due to these attributes, cinnamon has emerged as a promising candidate for controlling bacterial diseases in agriculture, offering a natural and sustainable alternative to chemical treatments.

Natural plant extracts, particularly cinnamon bark extract (CBE), are gaining attention as sustainable and cost-effective options for biocontrol agents. CBE offers a natural solution to manage BPB disease in rice, with potential efficacy against *B. glumae* [11]. Unlike synthetic chemicals, which can lead to heightened bacterial resistance and toxic accumulation in the environment, plant extracts offer a more natural and less harmful approach to disease management [9]. They present a low-risk alternative with minimal negative impact on both the environment and human health.

However, preserving the active compounds in CBE is a major challenge for researchers. These active compounds, especially the volatile ones, are susceptible to oxidation when exposed to air, which leads to the loss of their volatile and antibacterial properties [12]. In addition, some of the active compounds are likely to degrade in the body of the organism before they can exert their antibacterial effects. To address these stability and bioavailability challenges, nano formulations loaded with CBE and chitosan (CBE-CS) can be used. These nano formulations protect the active

ingredients from degradation and evaporation, thereby enhancing stability and reducing the required dosage. Herbal nanogels, which are often prepared with chitosan and other polymers, provide a stable and effective delivery system for these sensitive compounds. Encapsulating CBE in a nano formulation significantly protects its heat-sensitive components, making them less volatile. In addition, nanoencapsulation enhances the desirable properties of bioactive components by encapsulating them in a uniform matrix, which may improve their antibacterial activity [13]. This study successfully developed a nano formulation loaded with CBE to enhance the stability and effectiveness of cinnamon's active compounds in inhibiting *B. glumae*, the causative agent of BPB in rice fields. The research was conducted in three stages: first, the preparation and characterization of CBE through in vitro experiments to assess its efficiency and composition; second, the formulation of chitosan (CS)-based nano formulations with varying CBE concentrations and their impact on bacterial morphology; and finally, the evaluation of these nano formulations on infected rice plants in a controlled greenhouse environment. The findings demonstrate the potential of CBE-CS nano formulations as a promising biocontrol strategy for managing BPB in rice cultivation, offering improved compound stability and antibacterial efficacy.

## 2. Methodology

### 2.1 Materials and methods

The basic chemicals used in the laboratory experiments—including acetic acid (glacial, analytical grade), sodium tripolyphosphate (TPP), and Tween 80—were procured from Merck (Darmstadt, Germany). Chitosan (medium molecular weight, 75–85% deacetylated) was obtained from Sigma-Aldrich (St. Louis, MO, USA). All laboratory work was conducted in the Bacteriology Laboratories of the Department of Plant Protection, Faculty of Agriculture, Institute of Biological Sciences, Universiti Putra Malaysia, Selangor, Malaysia. Additionally, greenhouse facilities within the same faculty were utilized to evaluate the efficiency and effectiveness of the developed nano-extract across different germination models. The experiments were conducted with replications to ensure the reliability of the results. A completely randomized design (CRD) was employed for statistical analysis. One-way analysis of variance (ANOVA) was performed using SAS software (version 9.4) to evaluate differences among treatment groups. The least significant difference (LSD) test was applied at a 5% significance level ($P < 0.05$) to determine statistically significant differences between treatment means (Supporting Information).

### 2.2 Preparation of cinnamon bark extract

The cinnamon bark was washed, dried, and grounded into a fine powder using an electric grinder. 20g of the cinnamon bark powder was mixed with 200 ml of methanol solution in a 250 ml beaker. The beaker was placed on an orbital shaker for three days at room temperature. The mixture was then filtered using Whatman No. 1 filter paper, followed by bacterial filters. The filtered extracts were exposed to 60°C in a water bath for 30 minutes to evaporate the methanol. The filtered CBE were stored at 4°C until use [14].

### 2.3 Evaluation of CBE

#### 2.3.1 Antibacterial activity.
The antibacterial activity of the CBE against *B. glumae* was evaluated using zone of inhibition (ZOI), the minimum inhibitory concentration (MIC), and the minimum bactericidal concentration (MBC) as following in next sections.

#### I. Zone of inhibition

The agar disk diffusion method (Kirby-Bauer test) was used to assess the antibacterial activity of CBE against *B. glumae*, following minor modifications from [15]. KB agar plates were prepared and inoculated with *B. glumae* cultures diluted to an OD of 0.15 at 600 nm. Four wells (4 mm diameter) were filled with 50 µl of CBE at concentrations of 100, 50, 25, and 12.5 µg/ml. Positive (streptomycin, 0.15 mg/ml) and negative (distilled water) controls were included. Plates were incubated at

37 °C for 24 hours, and the zones of inhibition (ZOI) were measured to evaluate antibacterial activity [16]. The antibacterial effect was calculated using the inhibition formula (1):

$$Inhibition\ effect\ (\%) = \frac{extract\ inibition\ hold\ diameter\ (mm)}{inhibition\ zone\ of\ positive\ control\ (mm)} \times 100 \tag{1}$$

### II. Assessment of MIC and MBC

The MIC of CBE was determined using the serial two-fold dilution method in a 96-well microtiter plate, following the procedure described by [17]. CBE concentrations ranged from 50% to 0.19%, each prepared in 150 μL of sterile Mueller-Hinton broth (MHB). A suspension of *B. glumae* adjusted to an optical density of 0.15 at 600 nm (1 × 10^7 CFU/mL, 0.1 mL) was added to each well, except for the negative control (MHB only) and the positive control (MHB with 15 μg/mL streptomycin). Plates were incubated at 37°C for 24 hours, after which the MIC was defined as the lowest CBE concentration that visibly inhibited bacterial growth. For MBC determination, 10 μL from wells showing no visible growth was subcultured onto King's B agar plates using the drop plate method, as described in [18], and incubated at 37°C for 24 hours. The MBC was recorded as the lowest concentration at which no bacterial colonies were observed. All experiments were performed in triplicate [19].

### III. Bacteria growth

The relative population size log curve was used to illustrate the growth strength of bacteria and to determine antibacterial activity according to the method presented in [20]. A *B. glumae* suspension (1 × 10^7 CFU/mL, 0.1 mL) was added to Mueller-Hinton Broth (MHB) medium in a 100 mL culture flask. CBE was then added at two concentrations: 1.2 × MIC and 1 × MIC, to separate bacterial suspensions. Distilled water was also added in the same volume to balance the equation. The prepared culture broth was incubated at 37 °C for 24 hours, and then shaken in a rotary shaker at 180 rpm. Culture turbidity was measured every 2 hours by assessing the optical density at 600 nm (OD 600 nm) using a spectrophotometer for 24 hours. Each set of tests was repeated three times to determine the overall mean.

**2.3.2 Chemical characterization of CBE.** The CBE contains a range of bioactive compounds that require detailed chemical and structural analysis. To achieve this, various analytical techniques are applied. Gas chromatography-mass spectrometry (GC-MS) identifies and quantifies volatile compounds, while Fourier-transform infrared spectroscopy (FTIR) reveal functional groups.

### I. Gas chromatography-mass spectrometry

Chemicals in CBE were identified using GC-MS technology [21]. The GC-MS analysis was conducted following the method described by [22]. An SLB-5ms capillary column (30 m × 0.25 mm i.d. × 0.25 μm film thickness) was utilized. The initial temperature was set at 50 °C for 3 minutes, then increased at a rate of 10 °C/min until reaching 250 °C, where it was maintained for 10 minutes. The temperature was then raised to 300 °C and held for another 10 minutes. Mass spectra were recorded at 70 eV with a scan interval of 0.1 seconds, covering a mass range from 40 to 700 Da. The total analysis time was 45 minutes. Helium (99.999%) was used as the carrier gas at a flow rate of 0.8 mL/min. Chemical components were identified based on retention times, retention indices, and mass spectral data from the FFNSC1.3.lib, NIST11.lib, and WILEY229.LIB libraries, as well as relevant literature sources.

### II. Fourier-transform infrared spectroscopy

Infrared spectroscopic analysis was conducted using a FTIR. A small drop of the CBE sample was applied to the crystal of the FTIR spectrometer. The scanning range was set from 400 to 4000 cm^-1 with a resolution of 4 cm^-1. To ensure the accuracy of the results, data collection was repeated three times.

**2.3.3 Morphological characterization.** Three different types of microscopes were employed to characterize the morphology of the cinnamon bark extract (CBE), each providing distinct insights into its structural features.

### I. Scanning electron microscopy SEM

SEM was used to observe morphological changes in *B. glumae* treated with CBE, following [23]. Bacteria cultured on King's B agar were exposed to CBE at MIC, a streptomycin control, and distilled water (negative control). Inhibition zones were excised, fixed in 4% glutaraldehyde, and treated with osmium tetroxide before undergoing graded acetone dehydration. Samples were then dried, gold-coated, and examined under SEM.

### II. Transmission electron microscopy TEM

TEM was used to assess structural changes in *B. glumae* after CBE treatment, following [24]. Bacteria grown in Mueller-Hinton broth were treated with CBE at MIC, centrifuged, and fixed in 4% glutaraldehyde. After osmium tetroxide post-fixation, samples were dehydrated, embedded in resin, sectioned, and stained with uranyl acetate and lead citrate before TEM imaging [25].

### III. Confocal laser scanning microscopy CLSM

CLSM was used to generate high-resolution 3D images of *B. glumae* biofilms. Bacterial suspensions incubated with CBE at MIC were centrifuged, washed, and stained with SYTO 9 and propidium iodide. Stained samples were mounted on glass slides and imaged under CLSM to assess membrane damage [26].

### 2.4 CBE-loaded nano formulations

**2.4.1 Preparation of CBE-chitosan nano formulations.** A nano bactericide targeting *B. glumae* was prepared using chitosan loaded with CBE, following [27] with modifications. CBE was incorporated into chitosan nanoparticles using an oil-in-water emulsion and ionic gelation technique. Chitosan powder (≥75% deacetylation, Sigma Aldrich) was dissolved in 1% acetic acid and stirred overnight to form a 1% (w/v) CS solution. CBE was mixed with Tween 80 (1:1 v/v) and stirred for 1 hour until homogeneous. For nanoparticle formation, 5 ml of the CS solution was combined with the CBE-Tween 80 mixture and stirred until viscous. TPP (0–4% w/v) was then added to 15 ml of the CS mixture and stirred at 1000 rpm for 15 minutes. Finally, 5 ml of distilled water was added and left for 24 hours to complete the formulation. The resulting nano bactericidal formulation was designated CBE-CS. Fig 1 illustrates the preparation process.

**2.4.2 Evaluation of the developed nano-extract.** The efficacy of the developed chitosan-based nano formulation loaded with CBE was assessed through a series of in vitro experiments. These included antibacterial assays such as the zone of inhibition, MIC, and MBC, as described in previous sections. Additionally, two critical physicochemical parameters—particle size distribution and the encapsulation efficiency (EE%) along with loading capacity (LC%)—were evaluated to further characterize the nano formulation.

### I. Particle size analysis

The mean particle size (PS), polydispersity index (PDI), and zeta potential (ZP) of CBE-CS nanoparticles were analyzed using a Zeta sizer Nano ZS90 (Malvern, UK). PS and PDI were measured via dynamic light scattering (DLS) at 25°C, while ZP was determined based on electrophoretic mobility using the Helmholtz–Smoluchowski equation under a 40 V/cm electric field [28]. SEM (Thermo Scientific, Germany) was used to examine nanoparticle morphology after coating samples with a thin carbon layer. Stability was assessed by storing aqueous formulations at 25°C after 72 h, with PS, PDI, and ZP measured at the start and end of the storage period. All measurements were performed in triplicate.

### II. Encapsulation efficiency and loading capacity

Encapsulation efficiency (EE) and loading capacity (LC) were assessed to determine the effectiveness of the CB-CS nano bactericide formulation, following a modified method from [29]. Freeze-dried nano bactericide (400 mg) was mixed with 5 mL of 1 M HCl, vortexed for one minute, then combined with 2 mL of ethanol (99–100%) and vortexed again. The

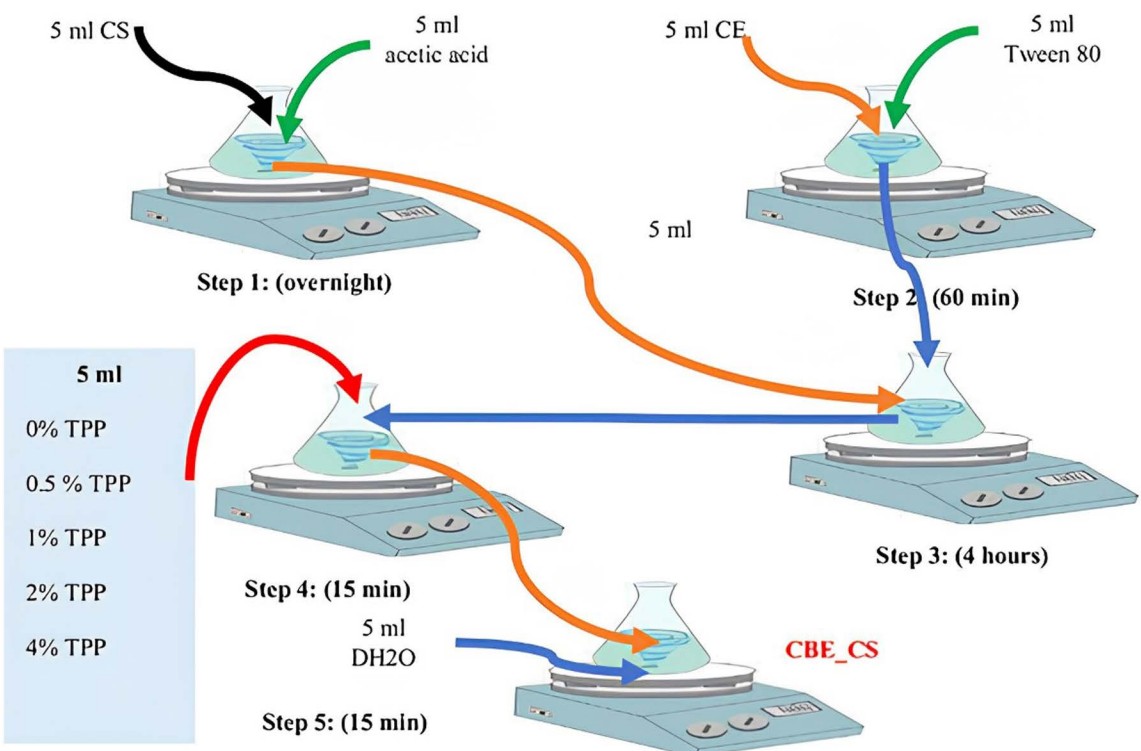

**Fig 1. Preparation of chitosan nanoparticles loaded with cinnamon bark extract using the ionic gelation method:** Step 1 – Chitosan (5 mL) was mixed with acetic acid (5 mL); Step 2 – Cinnamon extract (5 mL) was combined with Tween 80 (5 mL); Step 3 – The mixtures from Steps 1 and 2 were combined (1:1); Step 4 – Various concentrations of TPP were added to the resulting suspension; Step 5 – Distilled water (5 mL) was added to each formulation.

mixture was incubated at 60°C for 12 hours, centrifuged at 6000 rpm for 10 minutes, and the supernatant was analyzed using a UV-Vis spectrophotometer (Perkin Elmer, USA) at 200–400 nm. CBE concentration was quantified using a calibration curve ($R^2 = 0.9859$), with measurements performed in triplicate for accuracy. EE and LC were then calculated using standard formulas (2) and (3) [30].

$$EE\% = \frac{weight\ of\ CBE\ in\ the\ nanoparticles}{The\ initial\ weight\ of\ CBE\ in\ the\ system} \times 100 \tag{2}$$

$$LC\% = \frac{weight\ of\ CBE\ in\ the\ nanoparticles}{weight\ of\ nanoparticles} \times 100 \tag{3}$$

## 2.5 Assessment under greenhouse conditions

This section presents an evaluation of the efficacy of the CBE-CS formulation against BPB in rice, specifically under greenhouse conditions. The experimental design, including the inoculation process, treatment application, and controlled greenhouse conditions, is thoroughly detailed.

### 2.5.1 Germination characters under greenhouse conditions.
Surface-sterilized MR 219 rice seeds were soaked in a bacterial suspension containing activated *B. glumae communis* at a concentration of $1 \times 10^8$ CFU/mL for 4 hours. After

the inoculation, various concentrations of the CBE-CS formulation were applied to the inoculated rice seeds under the following treatment conditions:

- **GHSBT1**: Greenhouse-Seeds infected with *B. glumae* and treated with CBE-CS formulation at the MIC concentration of 12.5 µg/mL.

- **GHSBT2**: Greenhouse-Seeds infected with *B. glumae* and treated with CBE-CS formulation at twice the MIC (25 µg/mL).

- **GHSBT3**: Greenhouse-Seeds infected with *B. glumae* and treated with CBE-CS formulation at half the MIC (6.25 µg/mL).

- **GHSBS**: Greenhouse-Seeds infected with *B. glumae* and treated with a streptomycin formulation at a concentration of 15 µg/mL (positive control).

- **GHHS**: Greenhouse-Healthy seeds not inoculated with *B. glumae* and not treated with any bactericidal treatment (healthy control).

- **GHSB**: Greenhouse-Seeds infected with *B. glumae* and not treated with CBE-CS formulation (negative control).

Rice seeds were planted in circular plastic pots (30 cm diameter, 35 cm height) filled with clay soil in a greenhouse maintained at 30°C and 85–95% humidity. Four seeds per pot were planted in a randomized complete block design (RCBD) with three replicates. Plants were treated with solutions at 30, 45, and 60 days after sowing (DAS). Growth was assessed by measuring plant height 30 days after planting, and disease severity was monitored after inflorescence emergence. After 110 days, yield traits, including productive stems, seeds per inflorescence, 1000-seed weight, dry seed weight, and bud weights (fresh and dry), were measured. Plants were dried for two days before weighing.

**2.5.2 Evaluation of germination indicators.** The experiment was conducted in triplicate using a randomized complete block design (RCBD to evaluate the effectiveness of various CBE-CS formulation concentrations, as outlined below:

I. **Germination index and vigor index**

The germination percentage index and vigor index were calculated using (4) and (5) formulas [31].

$$Germination\ Index\ (\%) = \frac{Number\ of\ seeds\ germinated}{Total\ number\ of\ seeds} \times 100 \tag{4}$$

$$Vigour\ Index = Germination\ rate \times Total\ plant\ length \tag{5}$$

II. **Harvest index**

The harvest index was calculated as the ratio of total grain dry weight to total plant dry weight [32] as in equation (6). Filling and empty grains were sorted and weighed, with 1000 grains used for weight determination. The number of seeds per inflorescence was determined from nine samples, each containing three inflorescences, while spike numbers per cluster were calculated using 27 plants per treatment.

$$Harvest\ Index = \frac{Grain\ dry\ weight}{Total\ dry\ weight} \tag{6}$$

III. **Disease severity index**

The Disease Severity Index (DSI) provides a single value that summarizes the overall impact of a disease on an individual plant. In this study, the DSI for inflorescences treated with GHSBT1 to GHSB was monitored every two

days over a 90-day period until harvest. The severity of inflorescence disease was assessed using the standard visual rice evaluation system as described in [33]. Infected inflorescences, shoots, and seeds were categorized into ten levels based on the DSI, with infection rates ranging from 0% to over 80%: The DSI was calculated using equation (7) [34].

$$DIS = \frac{\sum (A \times B)}{\sum (N \times 9)} \times 100 \qquad (7)$$

where N is the total number of replications, A is the illness class (0–9), B is the number of plants per treatment indicating the disease class, and nine is a constant signifying the highest evaluation class.

### IV. Area under the disease progress curve

The Area Under the Disease Progress Curve (AUDPC) for a 90-day period ending on the day of harvest following planting (DAS) as seen in equation (8) [35].

$$AUDPC = \sum_{i=1}^{k} 1/2 \left( S_i + S_i - 1 \right) \times d \qquad (8)$$

Where $S_i$ is the disease severity at ith day, $K$ is the number of successive evaluations of disease, and $d$ is the interval between i and i-1 evaluation of disease.

### V. Protection index

In the same way, formula (9) was used to determine the protection index (PI) [36].

$$PI = \frac{a-b}{b} \times 100 \qquad (9)$$

Here, a represents the untreated control's AUDPC and b the treated control's AUDPC.

## 3. Results and discussion

The results were categorized into three main groups: (1) in vitro experiments conducted on the CBE, (2) evaluations of the developed nano-extract, and (3) greenhouse trials, as detailed below:

### 3.1 In vitro evaluation of CBE

#### 3.1.1 Antibacterial activity of CBE.

### I. Zone of inhibition, MIC and MBC

Cinnamon bark extract exhibited concentration-dependent antibacterial activity against *B. glumae* when dissolved in methanol. Higher CBE concentrations resulted in larger inhibition zones: 17.33 ± 0.33 mm (100%), 15.66 ± 0.33 mm (50%), 12.33 ± 0.33 mm (25%), and 10.66 ± 0.33 mm (12.5%). In comparison, the positive control (Streptomycin) produced a 20.00 ± 0.00 mm inhibition zone, while the negative control (Distilled Water) showed no activity (0.00 ± 0.00 mm). These results highlight CBE's potential as a natural antibacterial agent. Table 1 summarizes the inhibition zones across three replicates.

The MIC of CBE, determined using a two-fold dilution assay in 96-well microtiter plates, was 6.25 mg/mL, effectively inhibiting bacterial growth. Viable bacterial cells showed a red color, while non-viable cells showed no color change. The MBC, indicating bactericidal activity, was 12.5 mg/mL, as it completely eliminated bacterial viability (Fig 2). These results demonstrate CBE's potential as a potent antimicrobial agent under *in vitro* conditions.

**Table 1. Antibacterial activity of CBE methanol against _B. glumae._**

| Percentage (%) | 100% | 50% | 25% | 12.5% | Positive (+) Streptomycin (15 µg; ml) | Negative (-) Dist. H$_2$O |
|---|---|---|---|---|---|---|
| Inhibition diameter mm | 17.33 ± 0.33[b] | 15.66 ± 0.33[c] | 12.33 ± 0.33[d] | 10.66 ± 0.33[e] | 20.00 ± 0.00[z] | 0.00 ± 0.00[f] |

Mean n = 3.

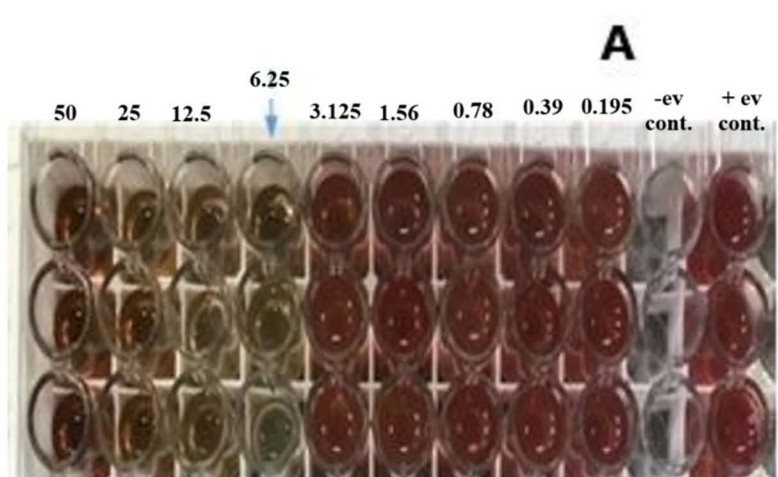 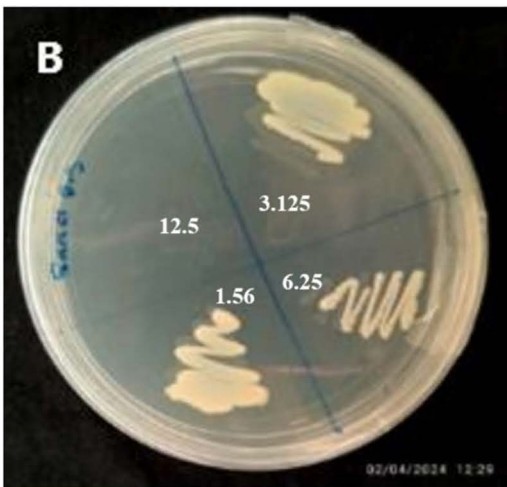

**Fig 2. Determination of MIC and MBC of the extract:** (a) MIC identified via serial dilution; color change from red to colorless at ≥6.25 mg/mL indicated growth inhibition. (b) MBC determined by subculturing non-turbid concentrations (6.25, 12.5, 25 mg/mL) on agar; 12.5 mg/mL was the lowest concentration showing no colony growth.

## II. Bacterial growth analysis

The growth rates of _B. glumae_ were studied under three conditions: untreated (control), treated with half the MIC concentration (0.5 MIC), and treated with the MIC concentration (1 MIC). Fig 3 illustrates the time course of _B. glumae_ growth. In the control group, optical density and growth rates increased steadily throughout the incubation period (2–18 hours). Treatment with 0.5 MIC of CBE resulted in a slight increase in optical density and growth rates, indicating partial inhibition of bacterial growth. However, treatment with 1 MIC of CBE significantly suppressed growth. Optical density decreased during the initial 0–4 hours of incubation and then stabilized, demonstrating the strong inhibitory effect of CBE on bacterial growth over an extended period.

### 3.1.2 Chemical composition analysis of CBE.

## I. GC-MS Analysis

GC-MS analysis of CBE identified 15 biologically active compounds, with _(Z)-3-phenylacetaldehyde_ (51.24%) being the most prevalent. Other significant compounds included _2-propenoic acid_, _cinnamaldehyde dimethyl acetal_, _hexadecanoic acid_, _eugenol, and oleic acid_, with smaller amounts of other compounds. The retention indices and times were compared with the FFNSC1.3.lib, NIST11 libraries, and previous studies [37] and [38]. The antibacterial properties of cinnamon are mainly attributed to compounds like _cinnamaldehyde_ and _eugenol_, which help reduce bacterial decomposition in food and cosmetics [39]. Table 2 shows the most common compounds in CBE.

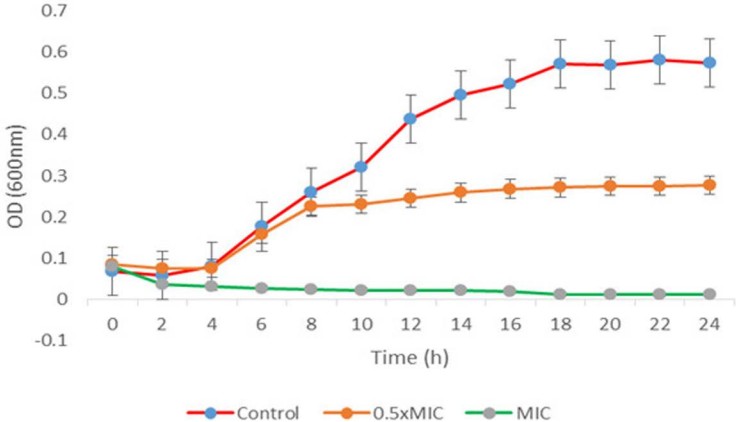

**Fig 3. The impact of CBE at MIC, 0.5 MIC, and in untreated control conditions on *B. glumae* was evaluated by measuring absorbance OD at 600 nm at two-hour intervals.** This allowed for a systematic observation of CBE's effects on bacterial growth dynamics over time.

**Table 2. Phytochemical constituents in CBE identified by GC-MS, comparing Retention Times (Rt), Retention Index (RI), Area %, molecular formula, mass spectral (MS), and Height Data with databases FFNSC1.3.lib, NIST11.lib, and WILEY229.lib.**

| No. | Chemical Component | Rt (min) | RI | Area (%) | Formula | MS | Height% |
|---|---|---|---|---|---|---|---|
| 1 | Cinnamaldehyde | 21.165 | 1218 | 0.58 | C9H8O | 373 | 1.08 |
| 2 | (Z)-3-Phenylacrylaldehyde | 23.707 | 1189 | 51.24 | C9H8O | 295 | 37.12 |
| 3 | Guaiacol | 25.582 | 1309 | 0.79 | C9 H10 O2 | 332 | 1.23 |
| 4 | Eugenol | 27.587 | 1392 | 3.13 | C10H12O2 | 398 | 5.14 |
| 5 | Coumarin | 28.723 | 1386 | 1.12 | C9 H8 O2 | 335 | 1.61 |
| 6 | Cinnamaldehyde dimethyl acetal | 29.477 | 1287 | 11.23 | C11H14O2 | 361 | 16.77 |
| 7 | 2-Propenoic acid, 3-(2-hydroxyphenyl) | 31.160 | 1577 | 13.24 | C9H8O3 | 343 | 14.15 |
| 8 | Muurolene | 33.938 | 1497 | 1.10 | C15 H24 | 404 | 1.60 |
| 9 | Eugenyl acetate | 34.917 | 1521 | 2.18 | C12 H14 O$_3$ | 388 | 3.21 |
| 10 | 2-Propenal, 3-(2-methoxyphenyl)- | 35.063 | 1378 | 2.99 | C10H10O$_2$ | 363 | 4.31 |
| 11 | 1,3-Benzenediol, 4-propyl | 37.942 | 1434 | 1.95 | C9H12O2 | 367 | 2.20 |
| 12 | Hexadecanoic acid | 51.155 | 1977 | 5.48 | C16 H32 O2 | 397 | 5.85 |
| 13 | 9-Octadecenoic acid (Z)-, methyl ester | 55.625 | 2085 | 0.68 | C19H36O2 | 383 | 1.08 |
| 14 | 10(E),12(Z)-Conjugated linoleic acid | 56.660 | 2183 | 2.06 | C18H32O2 | 371 | 2.06 |
| 15 | Oleic Acid | 56.845 | 2175 | 2.23 | C18H34O2 | 383 | 2.61 |

Fig 4 illustrates that CBE gas chromatography revealed 15 active substances.

II. FTIR analysis

FTIR analysis of CBE revealed key peaks that indicate the presence of various functional groups. The peak at 1727 cm$^{-1}$ corresponds to a carbonyl bond in aldehydes, while peaks at 1679 cm$^{-1}$ and 1626 cm$^{-1}$ are associated with aldehyde carbonyl groups, like those in cinnamaldehyde. The peak at 1573 cm$^{-1}$ reflects the C-skeletal vibration of the aromatic ring, linked to eugenol. Other notable peaks include 1450 cm$^{-1}$ for alcohol C–OH bending, 1294 cm$^{-1}$ for C–H bending in aromatic rings, and 1248 cm$^{-1}$ for ester C–O–C and C–OH stretching vibrations. Peaks at 973 cm$^{-1}$ and 748 cm$^{-1}$ are related to C–H bending and benzene ring vibrations, respectively. These peaks are characteristic of compounds like cinnamaldehyde and eugenol in cinnamon bark as depicted in Fig 5. Minor moisture content may slightly affect the signal, despite efforts to control moisture levels [40].

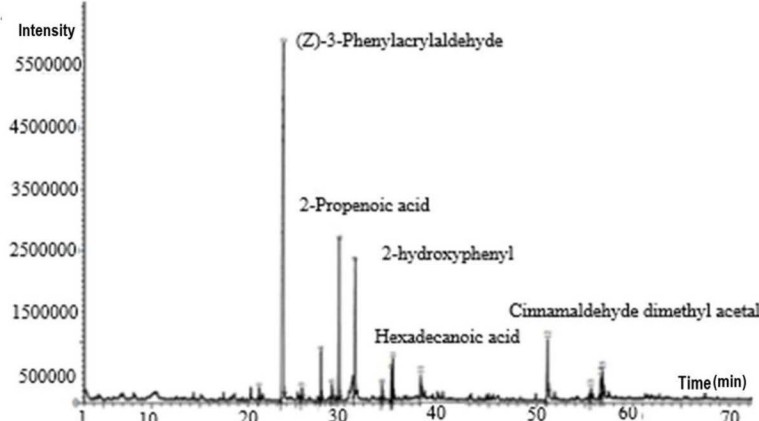

**Fig 4. GC-MS chromatogram of CBE, depicting phytochemical analysis characterization of CBE.**

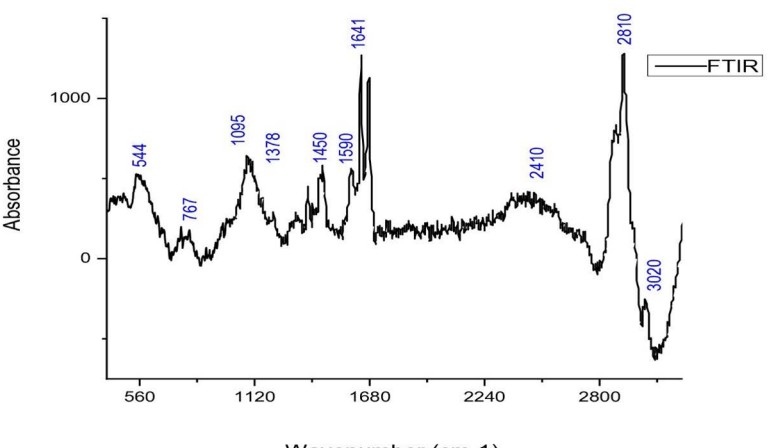

**Fig 5. Fourier Transform Infrared Spectra Analysis of Pure Cinnamon Bark Extract Highlighting Functional Groups and Chemical Composition.**

**3.1.3 Morphological characteristics.** The morphological and structural changes in *B. glumae* cells following treatment with CBE were analyzed using SEM, TEM, and CLSM, as illustrated in Fig 6.

I. SEM analysis

Fig 6(A) highlights the physical alterations in the cell walls of *B. glumae*. In Fig 6(A1), cells treated with distilled water (-ev. control) show intact, rod-shaped structures with smooth outer walls and no visible damage or cellular debris. In contrast, Fig 6(A2) reveals significant morphological disruptions in *B. glumae* treated with CBE at the MIC concentration for 24 hours, including visible damage and structural distortions in the cell walls. Fig 6(A3) shows the effects of streptomycin treatment (+ev. control), where the cell walls appear extensively damaged, collapsed, and torn. This destruction of the peptidoglycan layer led to severe cell wall compromise, causing cell detachment from the filter holder and eventual cell death.

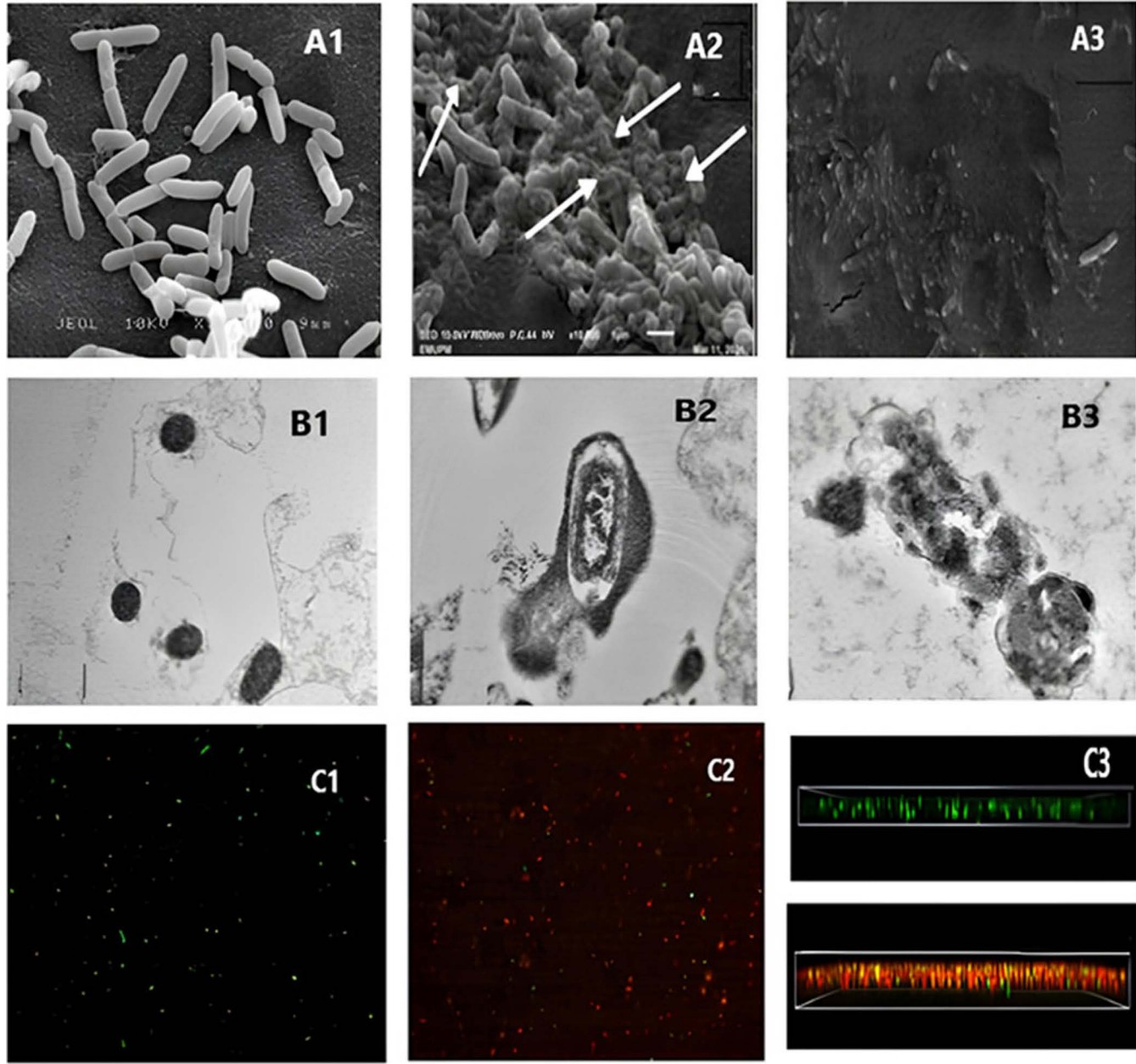

**Fig 6. SEM, TEM, and CLSM Analysis of *B. glumae* Cells After 24-Hour Treatment with CBE.** (A) SEM images (10,000×) showing intact bacterial cell walls with preserved cytoplasm in the distilled water control (A1), noticeable cell wall damage after CBE treatment (A2), and complete bacterial destruction with streptomycin(A3). (B) TEM images illustrating intact *B. glumae* cells in the control (B1), membrane disruption and cytoplasmic leakage after CBE treatment (B2), and complete bacterial destruction with streptomycin (B3). (C) CLSM images showing live green cells in the distilled water control (C1), increased dead cells (red/yellow) with CBE treatment at MIC (C2), and a 3D orthogonal biofilm structure of *B. glumae* treated with DH$_2$O and CBE (C3).

## II. TEM analysis

The integrity of the cytoplasmic membrane and internal morphology of *B. glumae* were examined using TEM, as shown in Fig 6(B). Fig 6(B1) displays cells treated with distilled water (-ve control) for 24 hours, showing uniform and intact

cell walls, evenly distributed cytoplasm, and distinct electron-dense structures inside the cells. In contrast, Fig 6(B2) demonstrates the effects of CBE treatment at the MIC concentration, revealing disruptions in the cytoplasmic membrane, indicating damage. Fig 6(B3) depicts cells treated with streptomycin (+ve. control), showing a collapsed and irregular cytoplasmic membrane with thickened cell walls. The severe membrane damage resulted in leakage of intracellular contents, compromising cellular integrity and viability.

III. **CLSM analysis**

The effects of CBE on *B. glumae* biofilms were further assessed using CLSM with LIVE/DEAD staining, as shown in Fig 6(C). Green fluorescence indicates live cells, while red fluorescence represents dead cells. Fig 6(C1) shows biofilms treated with distilled water (-ve. control), displaying a high density of live cells, evident from the abundance of green fluorescence. In contrast, Fig 6(C2) shows biofilms treated with CBE at the MIC concentration, where red fluorescence dominates, indicating a large number of dead cells. These results demonstrate the significant impact of CBE on the viability of *B. glumae* cells within biofilms, effectively reducing live cell populations and increasing cell death. Together, these analyses provide strong evidence of the antimicrobial and biofilm-disrupting properties of CBE against *B. glumae*. The cross section of the examined cells is shown in Fig 6(C3).

### 3.2 Characterization of CBE-loaded nanocomposites

**3.2.1 Morphology of nanoparticles.** Scanning electron microscopy was used to examine the morphology of nanoparticles at various TPP concentrations (0%, 0.5%, 1%, 2%, and 4%), as shown in Fig 7(A–E). TEM images revealed that the nanocomposite was nearly circular without TPP (Fig 7.2A). As TPP concentration increased (Fig 7.2B–D), the nanoparticles grew in size and exhibited a sharper, crystalline surface. The spherical nanoparticles had lighter edges, indicating TPP coating. Despite some particle clustering, the overall distribution of the extract remained homogeneous, with crystalline nanoparticle surfaces.

**3.2.2 Particle size, polydispersity index, and zeta potential analysis.** The physical properties of nanoparticles, including particle size, polydispersity index (PDI), and zeta potential (ZP), are critical in determining their behavior in biological environments. In this study, the PDI was calculated using light scattering, while the ZP was measured to evaluate particle stability. At 0% TPP, the particles were large (790.5 nm) with a high PDI (1), indicating instability. As TPP concentration increased to 0.5%, the particle size dramatically decreased to 43.66 nm with lower PDI of 0.288, suggesting more stable and uniform nanoparticles. However, further increases in TPP concentration led to a rise in particle size (51.72 nm at 1%, 79.82 nm at 2%, and 106.1 nm at 4%) and higher PDI values, indicating agglomeration and increased heterogeneity due to excessive cross-linking or TPP leakage. The results align with earlier studies, showing that PDI values below 0.45 indicate a narrow size distribution, as seen at 0.5% TPP. ZP values were positive, indicating protonation of chitosan's amino groups. A decrease in ZP with higher TPP concentrations suggests reduced positive charges on the particle surface, likely due to TPP interaction. All formulations met the required ZP threshold for stability. These findings are consistent with previous research, highlighting the importance of optimizing TPP concentration for nanoparticle stability. Table 3 displays the effects of different TPP concentrations on the NPs', PS, PDI, and ZP.

**3.2.3 Encapsulation efficiency and loading capacity.** The EE and LC of CBE-CS nano capsules were measured using UV-visible spectrophotometry. As TPP concentration increased, the loading capacity of the nano capsules improved, rising from 25.65% at 0% TPP to 33.9% at 4% TPP. This suggests that higher TPP levels enhance the ability of the nano capsules to incorporate CBE. However, excessive TPP can lead to a rigid, dense nanoparticle structure, reducing available space for loading and decreasing LC. Encapsulation efficiency remained stable across all TPP concentrations, ranging from 48.65% to 48.78%, indicating a potential saturation point beyond which further increases in TPP had little effect. These findings are consistent with previous studies. For instance, [29,30] and [41] also observed stable encapsulation efficiencies in similar formulations. Conversely, studies by [42] and [43] reported increases in both EE and

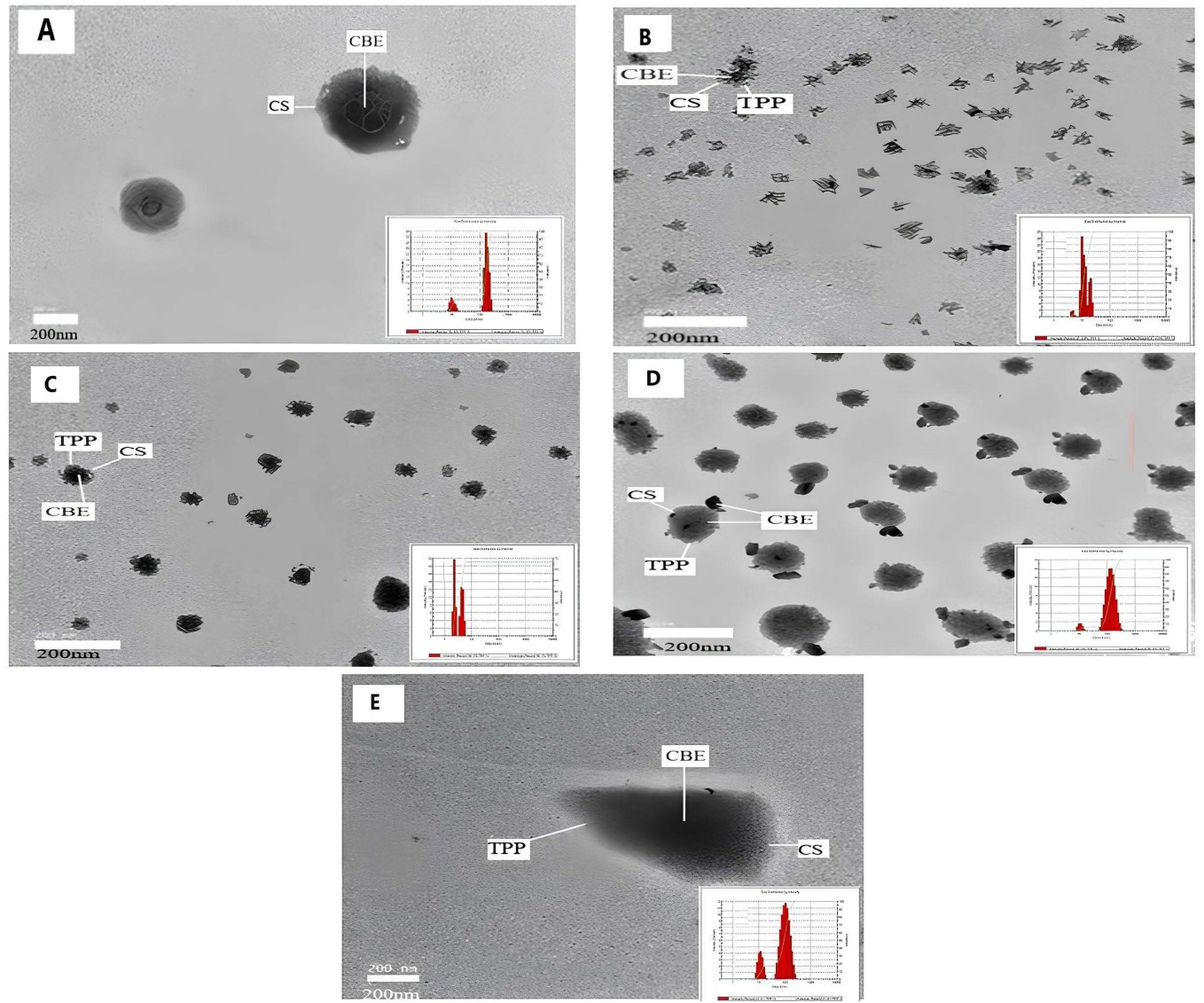

**Fig 7. TEM images of CBE-CS nano bactericide formulation with different concentrations of TPP:** (A) 0%, (B) 438 0.5% (C) 1%, (D) 2% and (E) 4% (scale bar is 200 nm).

**Table 3. The impact of TPP concentration on the CBE-CS nano bactericide formulation's average PS, PDI, and ZP.**

| Nano bactericide | Average size (d. nm) | PDI | Zeta Potential (mV) | Mob µmcm/Vs | Cond mS/cm |
|---|---|---|---|---|---|
| 0% TPP | 435.68 ± 22.14[a] | 1.00 ± 0.00[a] | 7.79 ± 0.04[a] | −0.47 ± 0.04[d] | 0.97 ± 0.02[a] |
| 0.5% TPP | 43.66 ± 0.23[a] | 0.28 ± 0.03[c] | 1.78 ± 0.04[b] | 0.61 ± 0.03[a] | 0.34 ± 0.02[c] |
| 1% TPP | 51.72 ± 0.04[a] | 0.57 ± 0.02[b] | 1.01 ± 0.01[b] | −0.20 ± 0.06[c] | 0.90 ± 0.03[a] |
| 2% TPP | 79.67 ± 0.17[a] | 0.68 ± 0.01[b] | −2.58 ± 0.08[c] | 0.13 ± 0.03[b] | 0.53 ± 0.07[b] |
| 4% TPP | 106.06 ± 0.42[a] | 0.96 ± 0.00[a] | −6.11 ± 0.61[d] | 0.07 ± 0.01[b] | 0.90 ± 0.02[a] |
| **Mean 1–30** | 1.08E + 04 | 0.661 | 0.396 | 0.03099 | 0.732 |
| **Std Dev** | 1.96E + 04 | 0.325 | 4.81 | 0.3767 | 0.258 |

LC with higher TPP concentrations. Fig 8 illustrates the effect of different TPP concentrations on the LC and EE of the nano capsules.

**3.2.4 Antibacterial activity, MIC, and MBC of CBE-CS.** The antibacterial activity, MIC, and MBC of CBE-CS nano-extract formulations were evaluated against *B. glumae*. The inhibition zone test showed varying antibacterial efficacy, with the DIZ increasing as TPP concentrations rose. The nano-extract with 0% TPP exhibited the smallest inhibition zone (8.6 mm), while formulations with 0.5% to 2% TPP had DZI values ranging from 11.3 to 11.6 mm, and the 4% TPP formulation showed 11.3 mm. The streptomycin control had the highest DZI at 20 mm. The highest PIRG percentage (58%) was observed in the 2% TPP formulation, while the lowest (43%) was in the 0% TPP formulation. The results, summarized in Table 4. The MIC was 15.6 µmol/mL, and the MBC was 31.25 µmol/mL, demonstrating the strong antibacterial potential of the CBE-CS nano-extract.

### 3.3 Greenhouse experiments

**3.3.1 Evaluation of germination indicators.** The effect of CBE-CS on rice seedling physiology was assessed after 14 days by measuring germination percentage, shoot and root length, vigor index, and disease severity

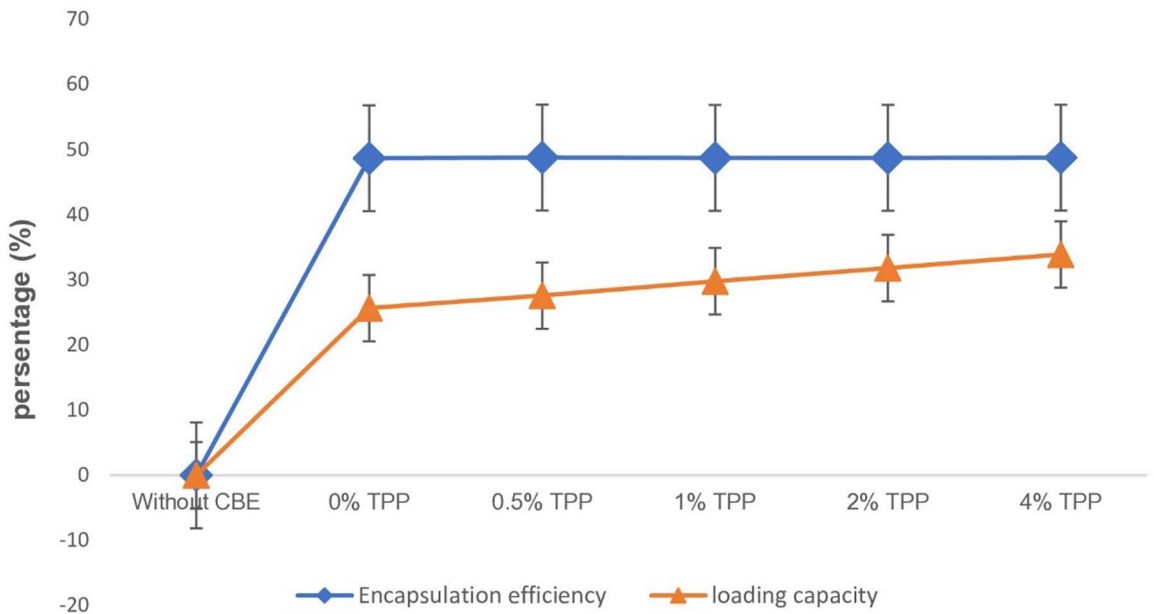

**Fig 8. Effect of CBE loading at different TPP concentrations on encapsulation efficiency and loading capacity (Value indicates mean of four replicates).**

**Table 4. Disc diffusion method-based antibacterial activities of CBE-CS nano bactericide formulation and the antibiotic streptomycin against *B. glumae*.**

|  | CBE nano bactericide formulation | | | | | Distilled water | Antibiotic Streptomycin |
|---|---|---|---|---|---|---|---|
|  | CBE-CS0 | CBE-CS0.5 | CBE-CS1 | CBE-CS2 | CBE-CS 4 |  |  |
| Diameter of zone of inhibition (mm) | 8.6± 0.33[c] | 11.3 ± 0.33[b] | 11.5 ± 0.28[b] | 11.6 ± 0.76 [b] | 11.3 ± 0.33[b] | 0.0 ± 0.0 [d] | 20 ± 0.0[a] |
| PIRG | 43% | 56.5% | 57.5% | 58.% | 56.5% | – | – |

**Table 5. Effects of CBE-CS on rice seedlings inoculated with *B. glumae*—Seed germination, Plant height, Root length, Seed vigor Index, and Disease severity under greenhouse conditions.**

| Treatment | Seed germination (%) | Shoot length (cm) | Root length (cm) | Seed vigor index | Disease severity |
|---|---|---|---|---|---|
| GHSBT1 | 53.30±5.77[c] | 10.39±1.19[a] | 5.25±1.03[a] | 892.89±123.09[dc] | 31.8±3.91[b] |
| GHSBT2 | 72.16±2.94[b] | 10.96±0.92[a] | 6.86±0.35[a] | 1261.10±165.9[abc] | 16.66±1.1[c] |
| GHSBT3 | 76.60±5.77[ab] | 11.18±0.65[a] | 6.35±0.52[a] | 1281.77±33.04[ab] | 15.57±0.63[c] |
| GHSBS +ev | 89.96±1.93[a] | 11.25±0.54[a] | 7.07±2.14[a] | 1527.12±77.48[a] | 9.33±1.13[c] |
| GHHS | 82.16±2.94[ab] | 10.01±0.91[a] | 3.85±0.63[a] | 1145.23±153.5[dc] | 00.0±0.0 |
| GHSB -ev | 46.63±5.07[c] | 9.90±1.73[a] | 5.58±0.86[a] | 787.08±113.6[d] | 39.3±4.0[a] |

Different letters in the same column differ significantly (P 0.05). Data are the averages over three replications.

(Table 5). Germination rates varied by treatment, with the positive control (GHSBS) showing the highest rate (89.96%), while treated models (GHSBT1, GHSBT2, GHSBT3, HS) recorded 53.30%, 72.16%, 76.6%, and 82.16%, respectively. The negative control (GHSB) had the lowest germination rate (46.63%). Shoot lengths ranged from 9.9 to 11.25 cm, and root lengths from 3.85 to 6.86 cm, showing similar growth patterns across treatments.

The seed vigor index varied across models, with the positive control (GHSBS) showing the highest value (1527.12), indicating strong seedling health. Treated models (SBT1, SBT2, SBT3) had lower vigor indices (892.89–1281.7), suggesting a potential impact of treatments. The negative control (GHSB) had the lowest index (787.08), highlighting pathogen-induced stress. Disease severity was lowest in the positive control (GHSBS) at 9.33, indicating a significant pathogenic effect.

After 130 days of cultivation, rice spikes were harvested, and key yield parameters were assessed, including the number of tillers, number of productive tillers, number of grains per panicle, grain filling percentage, and 1000-grain weight. These results are summarized in Table 6.

Yield parameters varied across treatments. Tillers per plant were highest in the healthy seed group (8.46) and lowest in the negative control (6.86), while treated groups (GHSBT1, GHSBT2, GHSBT3) ranged from 7.16 to 8.40. Productive tillers showed no significant differences, with values between 6.73 and 7.43, except for the negative control (5.96). Grain count per spike increased with treatment concentration, peaking at 230.20 for GHSBT3, while the healthy group had the highest (246.10) and the negative control the lowest (184.53). All treatments improved grain filling percentage over the negative control, with no significant differences in 1000-grain weight. CBE treatment enhanced yield components by suppressing disease.

**Table 6. The effect of nano bactericide formulation on yield components in a BPB-infected rice plant grown in a glasshouse.**

| Treatment | Number of Tillers | Number of productive Tillers | Number of Grains/Panicles | Filling % | 1000 grain weight (g) |
|---|---|---|---|---|---|
| GHSBT1 | 7.16±0.33[b] | 6.73±0.12[ab] | 206.03±0.27[e] | 61.06±032[e] | 23.52±0.00[c] |
| GHSBT2 | 7.53±0.14[b] | 7.13±0.03[a] | 222.33±0.55[d] | 65.60±0.12[d] | 23.66±0.00[b] |
| GHSBT3 | 8.40±0.30[a] | 7.43±0.12[a] | 230.20±0.75[c] | 68.53±0.12[d] | 23.72±0.01[ab] |
| GHSBS+ev | 8.33±0.08[a] | 6.86±0.46[ab] | 239.03±0.75[b] | 71.46±0.68[b] | 23.75±0.02[a] |
| GHSB-ev | 6.86±0.03a | 5.96±0.31[b] | 184.53±0.12[f] | 46.86±0.42[b] | 23.76±0.02[d] |
| GHSBH | 8.46±0.30[b] | 6.36±0.21[ab] | 246.10±0.36[a] | 76.63±0.38[f] | 23.41±0.17 [a] |

Different letters in the same column differ significantly (P 0.05). Data are the averages over six replications.

### 3.3.2 Assessment of DSI, AUDPC, DRI, and PI.

**I. Disease severity index**

Plant pathologists utilize the DSI with ordinal scales to generate data based on consecutive ranges, typically measured as the percentage of the symptomatic area present on specimens. The severity indices of BPB disease in rice seedlings were assessed over a 90–118-day period under greenhouse conditions. The progression of BPB disease is illustrated in Fig 9.

   In all models, disease severity indexes were markedly reduced and compared to the negative control group (GHSB). Specifically, the application of 7.5 µl/ml (GHSBT1), 15 µl/ml (GHSBT2), 30 µl/ml (GHSBT3), and the positive control (15 µg/ml, GHSBS) treatments resulted in significant reductions in DSI by approximately 78%, 67%, 41%, and 13%, respectively, after 108 days from sowing. In contrast, the negative control group experienced a disease severity of 87%. Among the treatments, the healthy plant group (GHHS) displayed no symptoms of BPB disease for the entire period between 90 and 108 days, underscoring the efficacy of preventive measures in disease management.

**II. AUDPC**

The most popular technique for estimating the AUDPC, known as the trapezoidal method, involves discretizing the time variable—days in this case—and figuring out the average illness severity that exists between each pair of consecutive time points as shown in Table 7. All treatments led to a significant reduction in the AUDPC index values. Specifically, the index values were 582, 699, 792, and 878 for study models GHSBS, GHSBT3, GHSBT2, and GHSBT1, respectively, in comparison to the negative control (GHSB), which had an index value of 979.

   The disease reduction index was calculated by averaging the treatment values over time, resulting in a total disease reduction of 100%. Notably, all concentrations significantly increased the disease reduction index compared to the negative control (GHSB). This suggests that the treatments were effective in slowing disease progression and supporting suppression by preventing pathogen invasion.

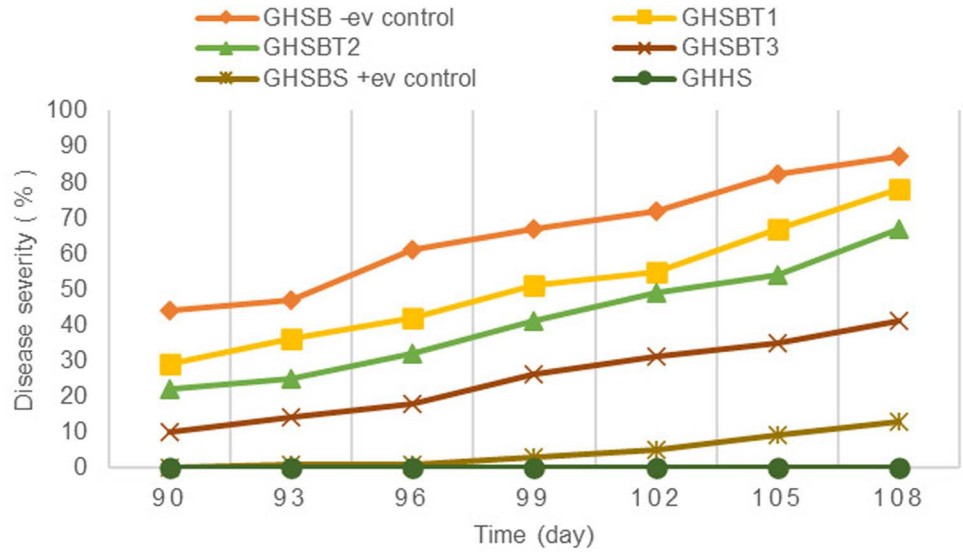

**Fig 9. Impact of CBE-CS formulations on BPB disease severity in rice seedlings 90 days after planting under greenhouse conditions.**

**Table 7. Efficacy of different CBE constrictions against BPB disease according to AUDPC index under greenhouse conditions.**

| Treatment | 1-30 Days | 31-60 Days | 61-90 Days | 90-118 Days | AUDPC |
|---|---|---|---|---|---|
| GHSBT1 | 71 | 142 | 294 | 371 | 878 |
| GHSBT2 | 63 | 137 | 228 | 364 | 792 |
| GHSBT3 | 52 | 126 | 207 | 314 | 699 |
| GHSBS +ev | 31 | 117 | 158 | 276 | 582 |
| GHSB -ev | 78 | 143 | 337 | 421 | 979 |

Different letters in the same column differ significantly (P 0.05). Data are the averages over six replications.

## III. DRI, and PI

In terms of the protection index, treatments GHSBT1, GHSBT2, GHSBT3, and GHSBS achieved values of 38.53%, 59.71%, 72.18%, and 81.34%, respectively, when compared to the negative control. These results indicate that all treatment concentrations were highly effective in controlling BPB. Fig 10 illustrates the values of both indices. Consequently, the treatments successfully hindered disease spread and contributed to disease suppression by preventing pathogen attacks.

## 4. Discussion

This study confirms the strong antibacterial potential of CBE against *B. glumae*, the pathogen responsible for BPB in rice. In vitro assays—ZOI, MIC, MBC, and bacterial growth curves—demonstrated a clear concentration-dependent inhibitory effect, aligning with previous reports [44] and [45]. The antimicrobial efficacy of CBE is attributed to its diverse phytochemical profile, particularly its volatile and lipophilic compounds such as cinnamaldehyde, (Z)-3-phenylacrylaldehyde, eugenol, geranial, and neral, as confirmed by GC-MS analysis [46] and [47]. These compounds disrupt bacterial membrane integrity, cause depolarization, and interfere with intracellular functions, ultimately

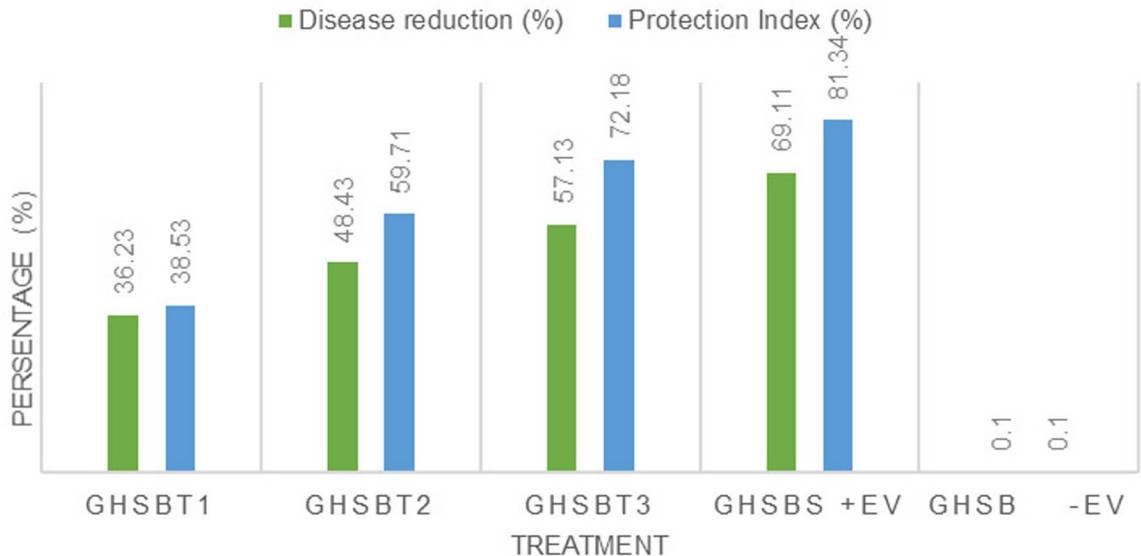

**Fig 10. Efficacy of different nano formulations against BPB disease according to disease index and protection index under greenhouse conditions.**

leading to cell death [48]. Microscopic observations revealed significant morphological damage in *B. glumae* cells treated with CBE, including membrane collapse and irregular shapes, indicating compromised structure and metabolism [49] and [50]. Confocal laser scanning microscopy further confirmed reduced viability and biofilm formation, emphasizing CBE's ability to prevent microbial colonization [51] and [22]. In addition, the phenolic compounds in CBE—especially flavonoids and phenolic acids—contribute antioxidant and radical-scavenging activity, which enhances antimicrobial effects through oxidative stress [52]. To improve the delivery and stability of CBE, chitosan-based nanoparticles were synthesized using sodium tripolyphosphate (TPP) as a cross-linker. Transmission electron microscopy (TEM) showed that 0.5% TPP yielded well-dispersed, spherical nanoparticles with reduced size. However, higher TPP concentrations led to particle agglomeration and size increase, likely due to gel network formation [53]. Increased TPP also caused a shift from monodisperse to polydisperse systems (higher PDI) and a transition from positive to negative zeta potential, attributed to the interaction between CBE's negatively charged functional groups and the nanoparticle surface [52]. Higher absolute zeta potential values indicated improved colloidal stability [54]. Greenhouse evaluations showed that all nanoformulations (7.5, 15, and 30 µL/mL) significantly enhanced seed germination and vigor compared to the negative control. Streptomycin produced the highest vigor index, followed closely by the 30 µL/mL CBE-CS treatment. These results suggest that the CBE-CS nanoparticles not only suppress *B. glumae* infection but also stimulate seedling growth, likely through both antibacterial activity and physiological enhancement [55]. The ability of nanoparticles to penetrate plant tissues and promote systemic effects further supports their dual protective and growth-promoting role.

Overall, the findings highlight the effectiveness of CBE-loaded chitosan nanoparticles as a natural, eco-friendly strategy for managing *B. glumae* infections and improving rice seedling health under controlled conditions

## 5. Conclusion

This study successfully developed a green nanobacterial formulation using CBE encapsulated within chitosan-tripolyphosphate (CS-TPP) nanoparticles to manage BPB effectively. GC-MS analysis of CBE identified 15 bioactive compounds, with (Z)-3-Phenylacrylaldehyde as the dominant constituent (51.24%), followed by 2-Propenoic acid, cinnamaldehyde dimethyl acetal, and other compounds known for strong antimicrobial properties. The antibacterial activity of CBE was confirmed through agar disk diffusion, MIC, and MBC assays, demonstrating a dose-dependent inhibition of *B. glumae*. MIC and MBC values were 6.25 µmol/mL and 12.5 µmol/mL, respectively. Microscopy techniques (SEM, TEM, and CLSM) revealed significant morphological alterations in *B. glumae* cells treated with CBE, such as membrane collapse and cytoplasmic leakage, confirming its bactericidal effects. CBE was then loaded into chitosan nanoparticles via ionic gelation using varying concentrations of TPP (0–4%). TEM imaging showed that the nanoparticles were generally spherical, with sizes ranging from 31 nm to 73 nm depending on TPP concentration. Notably, the CBE-CS 0.5 formulation had the smallest particle size, and the highest stability based on DLS, PDI, and zeta potential values. FTIR analyses confirmed successful encapsulation and molecular interactions among CBE, Chitosan, and TPP, indicating the structural integrity and crystallinity of the formulations. The antibacterial efficacy of the nanoformulations was further validated through inhibition zone assays and bacterial growth curves, with the 0.5 MIC, 1 MIC, and 2 MIC treatments (7.5, 15, and 30 µL/mL) effectively suppressing bacterial growth. Under greenhouse conditions, the CBE-CS formulations significantly controlled *B. glumae* infections without adversely affecting rice seed germination or seedling vigor.

In summary, the CBE-CS-TPP nanobacterial formulations demonstrated potent antibacterial activity, excellent physicochemical stability, and biosafety for use in rice cultivation. These findings highlight their potential as a sustainable, eco-friendly alternative to chemical pesticides, offering promising applications in crop protection and integrated disease management strategies to improve agricultural productivity.

## Supporting information

**S1 File. Supporting information.**
(DOCX)

## Acknowledgments

We express our sincere appreciation to Universiti Putra Malaysia (UPM) for its institutional support. We also gratefully acknowledge the Iraqi Ministry of Agriculture, represented by the Karbala Agriculture Directorate, for its moral support.

## Author contributions

**Conceptualization:** Qamar Mohammed-Naji, Dzarifah Zulperi, Erneeza Mohd Hata.

**Data curation:** Qamar Mohammed-Naji, Dzarifah Zulperi, Khairulmazmi Ahmad.

**Formal analysis:** Qamar Mohammed-Naji, Khairulmazmi Ahmad.

**Funding acquisition:** Qamar Mohammed-Naji, Khairulmazmi Ahmad.

**Methodology:** Qamar Mohammed-Naji, Khairulmazmi Ahmad.

**Project administration:** Qamar Mohammed-Naji.

**Resources:** Qamar Mohammed-Naji.

**Supervision:** Qamar Mohammed-Naji, Dzarifah Zulperi, Khairulmazmi Ahmad, Erneeza Mohd Hata.

**Validation:** Qamar Mohammed-Naji, Dzarifah Zulperi, Khairulmazmi Ahmad.

**Visualization:** Qamar Mohammed-Naji, Dzarifah Zulperi, Khairulmazmi Ahmad.

**Writing – original draft:** Qamar Mohammed-Naji, Dzarifah Zulperi, Erneeza Mohd Hata.

**Writing – review & editing:** Qamar Mohammed-Naji, Dzarifah Zulperi, Erneeza Mohd Hata.

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
