## [Decision Letter · Decision Letter 0]

PONE-D-25-07190Nano formulation development and antibacterial activity of cinnamon bark extract-chitosan composites against burkholderia glumae, the causative agent of bacterial panicle blight in ricePLOS ONE

Dear Dr. Alsutan,

Thank you for submitting your manuscript to PLOS ONE. After careful consideration, we feel that it has merit but does not fully meet PLOS ONE’s publication criteria as it currently stands. Therefore, we invite you to submit a revised version of the manuscript that addresses the points raised during the review process.

We look forward to receiving your revised manuscript.

Kind regards,

Abdelwahab Omri, Pharm B, Ph.D, Laurentian University

Academic Editor

PLOS ONE

**Journal Requirements:**

1. When submitting your revision, we need you to address these additional requirements. Please ensure that your manuscript meets PLOS ONE's style requirements, including those for file naming. The PLOS ONE style templates can be found at https://journals.plos.org/plosone/s/file?id=wjVg/PLOSOne_formatting_sample_main_body.pdf and https://journals.plos.org/plosone/s/file?id=ba62/PLOSOne_formatting_sample_title_authors_affiliations.pdf 2. Thank you for stating the following in the Acknowledgments Section of your manuscript: We thank the financial support provided by University of Putra Malaysia UPM. We also thank the Iraqi Ministry of Agriculture represented by the Karbala Agriculture Directorate for its moral support. We note that you have provided funding information that is not currently declared in your Funding Statement. However, funding information should not appear in the Acknowledgments section or other areas of your manuscript. We will only publish funding information present in the Funding Statement section of the online submission form. Please remove any funding-related text from the manuscript and let us know how you would like to update your Funding Statement. Currently, your Funding Statement reads as follows: The author(s) received no specific funding for this work.  Please include your amended statements within your cover letter; we will change the online submission form on your behalf. 3. We note that your Data Availability Statement is currently as follows: All relevant data are within the manuscript and its Supporting Information files. Please confirm at this time whether or not your submission contains all raw data required to replicate the results of your study. Authors must share the “minimal data set” for their submission. PLOS defines the minimal data set to consist of the data required to replicate all study findings reported in the article, as well as related metadata and methods (https://journals.plos.org/plosone/s/data-availability#loc-minimal-data-set-definition). For example, authors should submit the following data: - The values behind the means, standard deviations and other measures reported;- The values used to build graphs;- The points extracted from images for analysis. Authors do not need to submit their entire data set if only a portion of the data was used in the reported study. If your submission does not contain these data, please either upload them as Supporting Information files or deposit them to a stable, public repository and provide us with the relevant URLs, DOIs, or accession numbers. For a list of recommended repositories, please see https://journals.plos.org/plosone/s/recommended-repositories. If there are ethical or legal restrictions on sharing a de-identified data set, please explain them in detail (e.g., data contain potentially sensitive information, data are owned by a third-party organization, etc.) and who has imposed them (e.g., an ethics committee). Please also provide contact information for a data access committee, ethics committee, or other institutional body to which data requests may be sent. If data are owned by a third party, please indicate how others may request data access. 4. PLOS requires an ORCID iD for the corresponding author in Editorial Manager on papers submitted after December 6th, 2016. Please ensure that you have an ORCID iD and that it is validated in Editorial Manager. To do this, go to ‘Update my Information’ (in the upper left-hand corner of the main menu), and click on the Fetch/Validate link next to the ORCID field. This will take you to the ORCID site and allow you to create a new iD or authenticate a pre-existing iD in Editorial Manager. 5. Please upload a new copy of Figure 7 as the detail is not clear. Please follow the link for more information: https://blogs.plos.org/plos/2019/06/looking-good-tips-for-creating-your-plos-figures-graphics/
https://blogs.plos.org/plos/2019/06/looking-good-tips-for-creating-your-plos-figures-graphics/

Reviewers' comments:

Reviewer's Responses to Questions

**Comments to the Author**

1. Is the manuscript technically sound, and do the data support the conclusions?

Reviewer #1: Partly

Reviewer #2: Yes

Reviewer #3: Partly

2. Has the statistical analysis been performed appropriately and rigorously? 

Reviewer #1: I Don't Know

Reviewer #2: Yes

Reviewer #3: No

3. Have the authors made all data underlying the findings in their manuscript fully available?

Reviewer #1: Yes

Reviewer #2: Yes

Reviewer #3: Yes

4. Is the manuscript presented in an intelligible fashion and written in standard English?

Reviewer #1: No

Reviewer #2: Yes

Reviewer #3: No

5. Review Comments to the Author

**Reviewer #1:**  The manuscript in its current form is not readable and requires significant revisions. There are several issues related to readability, formatting, and scientific accuracy that need to be addressed before it can be considered for publication. The scientific content also appears to have several inaccuracies that must be corrected to ensure the integrity of the research presented. This reviewer strongly recommends that the authors refer to well-written, published scientific manuscripts for guidance on improving both the quality and readability of this work followed by thorough proof reading before submission.

Specific Comments:

Writing and Readability:

1. The manuscript is poorly written with several grammatical and typographical errors. The flow of ideas is difficult to follow, and the writing lacks cohesion.

2. Title Formatting: The title contains italics, which are unnecessary. Italics should only be used for bacterial names, and even in this case, the first letter of the genus should be capitalized, and no period is needed when writing the full genus name.

3. Figure Legends need to be written in a standard format. The current legends are not informative and lack essential details. For example, Figure 2 does not specify what the other wells represent after 3.125, and there are no labels for 2B. Additionally, Figure 1 lacks any kind of legend or explanation. Clear and concise figure legends are critical to help the reader interpret the figures properly.

4. Inconsistent Sections: In Line 312, the manuscript moves into the "Results and Discussion" section, but Line 313 continues discussing statistical analysis without the appropriate title. This is confusing and suggests lack of attention to important details while writing the manuscript.

5. The aspect ratio of the figures in Figure 7 should be consistent. The figures appear stretched or distorted, which impacts their clarity. The authors should ensure that all figures are presented with a consistent aspect ratio and that they are visually clear.

Scientific Errors:

There appear to be several scientific inaccuracies within the manuscript. For ex: It is unclear why the positive control used in Figure 2 was not also used in the greenhouse studies. Using streptomycin in plant studies would strengthen the data.

**Reviewer #2:**  chitosan composites against Burkholderia glumae, the causative agent of bacterial panicle blight in rice" presents an interesting and relevant approach to developing natural antimicrobial formulations for agricultural pathogens. The use of cinnamon bark extract combined with chitosan for nanoformulation and its evaluation against B. glumae holds promise for sustainable disease management in rice

**Reviewer #3: ** • Keywords: add chitosan to keywords

• Section 2. Material and method: (lines 104-109) is written poorly, containing un informative sentences. Chitosan and all chemicals are bought from local market with no specific brand and manufacturing company? This may lead to a non-reliable and non-repeatable results.

• Needs English language corrections. e.g. line 117 filtered s correct instead of filters.

• What is “inhibitory zone of inhibition”? (L 121) and (L123)

• “Bacteria growth evaluate” authors mean Bacterial growth evaluation???? (L142)

• In section II. Assessment of MIC and MBC, how much microorganism suspension was added?

• Sentences like “The resulting spectra and peaks were observed on a computer screen” are not necessary and can be deleted. (L 173)

• Section 2.2.1, results are presented in method section.

• Section 2.2.2, duration of stability studies is not mentioned.

• What was the λmax for EE and LC calculation? Not a range (200-400)

• Just one disk diffusion or MIC and MBC determination section was enough for both extract and nano formulation. These sections are repeated in method section for extract and nano formulation separately.

• Statistical analysis should be mentioned in method part, not in result section.

• FTIR spectra should be provided for both CBE and CBE chitosan nano-formulation.

• Figures need more quality and resolutions. Fig 2 is not informative and can be omitted.

• The writing in section 2 and 3 should be completely restructured.

• Discussion is poorly written with no discussing about the results of other studies.

• Conclusion: There is no need to explain about method in conclusion. Add the limitations.

6. PLOS authors have the option to publish the peer review history of their article (what does this mean? ). If published, this will include your full peer review and any attached files.

**Do you want your identity to be public for this peer review?** For information about this choice, including consent withdrawal, please see our Privacy Policy .

Reviewer #1: No

Reviewer #2: No

Reviewer #3: **Yes: ** Zahra Hesari

---

## [Author Response · Author response to Decision Letter 1]

21 May 2025

Response to Reviewers

Dear Editor,

We would like to express our sincere gratitude for the valuable feedback provided by the reviewers on our manuscript entitled: “Nanoformulation development and antibacterial activity of cinnamon bark extract-chitosan composites against Burkholderia glumae, the causative agent of bacterial panicle blight in rice” (Manuscript ID: PONE-D-25-07190).

We have carefully considered all the comments and suggestions provided by the reviewers and have revised the manuscript accordingly. Detailed responses to each comment have been provided in the revised version, and all necessary modifications have been made to improve the clarity, methodology, and overall quality of the manuscript.

We believe that the revised version now addresses all the concerns raised and hope that it meets the publication standards of PLOS ONE.

Thank you again for the opportunity to revise our work. We look forward to your positive response.

Journal Requirements:

1 Please ensure that your manuscript meets PLOS ONE's style requirements, including those for file naming.

Author Procedures// In response to the reviewer’s comment, we have ensured that the manuscript has been thoroughly updated to meet the PLOS ONE style requirements. line// All manus.

2 We note that you have provided funding information that is not currently declared in your Funding Statement. However, funding information should not appear in the Acknowledgments section or other areas of your manuscript.

Author Procedures// We have edited the Acknowledgment section to exclude financial disclosures, ensuring alignment with PLOS ONE’s requirements. line // 644-646

3 Please confirm at this time whether or not your submission contains all raw data required to replicate the results of your study. Authors must share the “minimal data set” for their submission.

Author Procedures// All raw data required to reproduce the findings of this study have been included in the submission. line// All

4 PLOS requires an ORCID iD for the corresponding author in Editorial Manager on papers submitted after December6th, 2016.

Author Procedures// The ORCID iD of the corresponding author has been provided. Line //16

5 Please upload a new copy of Figure 7 as the detail is not clear.

Author Procedures// A new copy of Figure 7 has been uploaded with clear detail. Fig 7

Review Comments

Reviewer 1

1 The manuscript is poorly written with several grammatical and typographical errors. The flow of ideas is difficult to follow, and the writing lacks cohesion.

Author Procedures// Many grammatical and typographical errors in the manuscript have been corrected as much as possible, and an attempt has been made to reconstruct the sequence of ideas and increase the coherence of the text. All lines

2 Title Formatting: The title contains italics, which are unnecessary. Italics should only be used for bacterial names, and even in this case, the first letter of the genus should be capitalized, and no period is needed when writing the full genus name.

Author Procedures// The title has been reformatted to use italics only for bacterial names, capitalize the first letter of the genus name, and remove the full period from the genus name. line //2-4

3 Figure Legends need to be written in a standard format. The current legends are not informative and lack essential details. For example, Figure 2 does not specify what the other wells represent after 3.125, and there are no labels for 2B.Additionally, Figure 1 lacks any kind of legend or explanation. Clear and concise figure legends are critical to help the reader interpret the figures properly.

Author Procedures// The figure descriptions have been rewritten to address valuable reviewer comments.

The title of Figure 2 has been revised to reflect what other wells represent. line// 336-339

In addition, Figure 1 has been reinterpreted. line// 216-220

4 Inconsistent Sections: In Line 312, the manuscript moves into the "Results and Discussion" section, but Line 313continues discussing statistical analysis without the appropriate title. This is confusing and suggests lack of attention to important details while writing the manuscript.

Author Procedures// The paragraph after the results and discussion section was reformulated to show the type of procedures followed and the statistical programs used to obtain the results and moved to methodology section. line// 113-117

5 The aspect ratio of the figures in Figure 7 should be consistent. The figures appear stretched or distorted, which impacts their clarity. The authors should ensure that all figures are presented with a consistent aspect ratio and that they are visually clear.

Author Procedures// Due to the large volume of results obtained, and to present the most relevant findings, multiple microscopic images were combined in Figure 7. This combination slightly affected the image clarity and accuracy. Nevertheless, the images effectively illustrated the key morphological changes in cell shape. Efforts were made to adjust the image dimensions and incorporate higher-quality, more accurate images in the final version of Figure 7 Fig 7

6 Scientific Errors: There appear to be several scientific inaccuracies within the manuscript. For ex: It is unclear why the positive control used in Figure 2 was also not used in the greenhouse studies. Using streptomycin in plant studies would strengthen the data.

Author Procedures// In this research and the associated greenhouse experiments, six study models were utilized, as outlined in Section 3.2.1. Three of the models were treated with varying concentrations of the developed nano-extract. The GHSBS model served as the positive control, while the GHSB model represented the negative control. Data from all six models were included to provide a comprehensive analysis and strengthen the overall findings line// 264-269

Reviewer 2

1 chitosan composites against Burkholderia glumae, the causative agent of bacterial panicle blight in rice"presents an interesting and relevant approach to developing natural antimicrobial formulations for agricultural pathogens.The use of cinnamon bark extract combined with chitosan for nanoformulation and its evaluation against B. glumae holdspromise for sustainable disease management in rice

Author Procedures// Thank you for your positive and encouraging feedback. We appreciate your recognition of the relevance and potential impact of our work. The combination of cinnamon bark extract with chitosan in a nanoformulated composite was indeed selected to explore an environmentally friendly and sustainable strategy for managing Burkholderia glumae, a significant pathogen in rice cultivation. We are encouraged that the reviewer finds this approach promising for agricultural disease control, and we hope our findings contribute meaningfully to the development of natural alternatives in plant protection. All

Reviewer 3

1 Keywords: add chitosan to keywords

Author Procedures// Chitosan has been added to keywords line //44

2 Section 2. Material and method: (lines 104-109) is written poorly, containing uninformative sentences. Are Chitosan and all chemicals bought from local market with no specific brand and manufacturing company? This may lead to non-reliable and non-repeatable results. Author Procedures// We acknowledge the concern regarding the clarity and specificity of the information provided in this section. To address this, we have revised the relevant sentences to include the source, brand, and manufacturer details of the materials used. This will enhance the reproducibility and reliability of the study. line// 105-112

3 Needs English language corrections. e.g. line 117 filtered s correct instead of filters.

Author Procedures// The world filters has been corrected to filtered. line// 123

4 What is “inhibitory zone of inhibition”? (L 121) and (L123)

Author Procedures// Thank you for your valuable feedback. It has been modified to the "zone of inhibition” line// 128 & 130

5 “Bacteria growth evaluate” authors mean Bacterial growth evaluation???? (L142)

Author Procedures// Thank you for your valuable feedback. It has been removed. The title became " Bacterial growth” line 152

6 In section II. Assessment of MIC and MBC, how much microorganism suspension was added?

Author Procedures// The paragraph was modified to read” A suspension of B. glumae adjusted to an optical density of 0.15 at 600 nm (~1 × 10⁶ CFU/mL) was added to each well (10 μL)” line// 144

7 Sentences like “The resulting spectra and peaks were observed on a computer screen” are not necessary and can be deleted. (L 173) Author Procedures// The sentence “The resulting spectra and peaks were observed on a computer screen” has been deleted. line// 186

8 Section 2.2.1, results are presented in method section

Author Procedures// The sentence “forming chitosan nanoparticles (200–1000 nm, zeta potential 20–60 mV)” has been deleted from section 2.2. line// 124

9 Section 2.2.2, duration of stability studies is not mentioned.

Author Procedures// In our study, the duration of stability studies in particle size analysis was 72 hours, and this was added to Section I- Particle size analysis. line //235

10 What was the λmax for EE and LC calculation? Not a range (200-400)

Author Procedures// λmax is the maximum absorbance wavelength of bioactive compound (e.g., cinnamaldehyde in CBE). It is determined by scanning the extract in a UV-Vis spectrophotometer, and its range typically between 200–600 nm. In CBE it is between ~290–300 nm, Cinnamaldehyde (major component) ~290–295, Phenolic compounds ~270–320. line// 243-244

11 Just one disk diffusion or MIC and MBC determination section was enough for both extract and nano formulation. These sections are repeated in the method section for extract and nano formulation separately.

Author Procedures// Valuable note, section 2.2.4 has been deleted. The equation for calculating the antibacterial effect using the inhibition formula (1) has been added to Section 2.1.2. line// 137-138

12 Statistical analysis should be mentioned in the method part, not in the result section.

Author Procedures// In response to the comments from Reviewers 1 and 3, the Statistical Analysis section has been revised to provide a clearer and more detailed explanation of the data analysis methods and calculation procedures used, ensuring alignment with the reviewers' suggestions and enhancing the overall clarity of the methodology. line// 113-117

13 FTIR spectra should be provided for both CBE and CBE chitosan nano-formulation.

Author Procedures// The FTIR spectral results for both CBE and CBE-CS were obtained; however, due to data redundancy and the extensive volume of results, which could impact on the overall length and focus of the manuscript, only the FTIR analysis of CBE is presented in this paper. The FTIR data for CBE-CS is available from the authors upon request. All

14 Figures need more quality and resolutions. Fig 2 is not informative and can be omitted

Author Procedures// The accuracy and clarity of the figures have been adjusted as much as possible, and Figure 2 has been adjusted. All figs.

15 The writing in section 2 and 3 should be completely restructured.

Author Procedures// Sections 2 and 3 have been completely restructured. Sec. 2 & 3

16 Discussion is poorly written with no discussion about the results of other studies

Author Procedures// The discussion section has been rewritten, considering the results of other studies. line 578-614

17 Conclusion: There is no need to explain about method in conclusion. Add the limitations.

Author Procedures// The conclusion section has been rewritten, considering the limitation. line// 616-640

---

## [Editor Report · Decision Letter 1]

Nano formulation development and antibacterial activity of cinnamon bark extract-chitosan composites against Burkholderia Glumae the causative agent of Bacterial Panicle Blight in rice

PONE-D-25-07190R1

Dear Dr. Qamar Mohammed Alsutan,

We’re pleased to inform you that your manuscript has been judged scientifically suitable for publication and will be formally accepted for publication once it meets all outstanding technical requirements.

Kind regards,

Abdelwahab Omri, Pharm B, Ph.D, Laurentian University

Academic Editor

PLOS ONE

---

## [Editor Report · Acceptance letter]

PONE-D-25-07190R1

PLOS ONE

Dear Dr. Mohammed-Naji,

I'm pleased to inform you that your manuscript has been deemed suitable for publication in PLOS ONE. Congratulations! Your manuscript is now being handed over to our production team.

Kind regards,

on behalf of

Dr. Abdelwahab Omri

Academic Editor

PLOS ONE